# Temperature sensitivity of dark $CO_2$ fixation in temperate forest soils

Rachael Akinyede[1,2], Martin Taubert[1], Marion Schrumpf[2], Susan Trumbore[2], Kirsten Küsel[1,3]

[1]Aquatic Geomicrobiology, Institute of Biodiversity, Friedrich Schiller University Jena, Dornburger Str. 159, 07743 Jena, Germany
[2]Department for Biogeochemical Processes, Max Planck Institute for Biogeochemistry, Hans-Knöll Str. 10, 07745 Jena, Germany
[3]German Centre for Integrative Biodiversity Research (iDiv) Halle-Jena-Leipzig, Puschstraße 4, 04103 Leipzig, Germany

*Correspondence to*: Kirsten Küsel (kirsten.kuesel@uni-jena.de)

**Abstract.** Globally, soil temperature to 1 m depth is predicted to be up to 4 °C warmer by the end of this century, with pronounced effects expected in temperate forest regions. Increased soil temperatures will potentially increase the release of $CO_2$ from temperate forest soils, resulting in important positive feedback on climate change. Dark $CO_2$ fixation by microbes can recycle some of the released soil $CO_2$, and $CO_2$ fixation rates are reported to increase under higher temperatures. However, research on the influence of temperature on dark $CO_2$ fixation rates, particularly in comparison to the temperature sensitivity of respiration in soils of temperate forest regions is missing. To determine the temperature sensitivity ($Q_{10}$) of dark $CO_2$ fixation and respiration rates, we investigated soil profiles to 1 m depth from beech (deciduous) and spruce (coniferous) forest plots of the Hummelshain forest, Germany. We used $^{13}C$-$CO_2$ labelling and incubations of soils at 4 and 14°C to determine $CO_2$ fixation and net soil respiration rates and derived the $Q_{10}$ values for both processes with depth. The average $Q_{10}$ for dark $CO_2$ fixation rates normalized to soil dry weight was 2.07 for beech and spruce profiles, and this was lower than the measured average $Q_{10}$ of net soil respiration rates with ~2.98. Assuming these $Q_{10}$ values, we extrapolated that net soil respiration might increase 1.16 times more than $CO_2$ fixation under a projected 4 °C warming. In the beech soil, a proportionally larger fraction of the label $CO_2$ was fixed into soil organic carbon than into microbial biomass compared to the spruce soil. This suggests primarily higher rate of microbial residue formation (i.e. turnover as necromass or release of extracellular products). Despite a similar abundance of the total bacterial community in the beech and spruce soils, the beech soil also had a lower abundance of autotrophs, implying a higher proportion of heterotrophs when compared to the spruce soil, hence might partly explain the higher rate of microbial residue formation in the beech soil. Furthermore, higher temperatures in general lead to higher microbial residues formed in both soils. Our findings suggest that in temperate forest soils, $CO_2$ fixation might be less responsive to future warming than net soil respiration and could likely recycle less $CO_2$ respired from temperate forest soils in the future than it does now.

**Keywords:** Dark $CO_2$ fixation; Respiration; Temperature sensitivity ($Q_{10}$); Microbial biomass carbon (MBC); Soil organic carbon (SOC); Temperate Forest soil.

## 1 Introduction

Most of Earth's terrestrial carbon stock is found in soils, with ~36 % occurring in the top 1 m depth of forest soils (Jobbágy and Jackson, 2000) based on the new carbon inventory of the global soil carbon pool (Hugelius et al., 2014; Schuur et al., 2015). Decomposition of soil organic carbon (SOC) provides one of the largest sources

of carbon dioxide ($CO_2$) to the atmosphere (Rastogi et al., 2002; Lal, 2004). Microbes can refix 3 – 6 % of $CO_2$ in temperate forest mineral soils before its release to the atmosphere (Akinyede et al., 2020; Spohn et al., 2019), through so-called dark $CO_2$ fixation (Miltner et al., 2005; Šantrůčková et al., 2018). Dark $CO_2$ fixation in soils is mediated by chemolithoautotrophic bacteria, largely via the Calvin Benson Bassham (CBB) pathway (Niederberger et al., 2015; Wu et al., 2014), the Wood Ljungdahl pathway (WLP), or the reverse tricarboxylic acid (rTCA) pathway (Beulig et al., 2016; Liu et al., 2018). Heterotrophic bacteria can also contribute to dark $CO_2$ fixation via anaplerotic carboxylation reactions associated with central microbial metabolism (Erb, 2011). The genetic potential for both autotrophic and heterotrophic $CO_2$ fixation has been demonstrated in various soils (Miltner et al., 2005; Šantrůčková et al., 2018) including temperate forest soils (Akinyede et al., 2022a, 2020; Kaiser et al., 2016).

The biomass of microbial communities serves as the entry point of carbon fixed from $CO_2$ into SOC, which includes both the intact microbial biomass carbon (MBC) pool and released microbial residues (Miltner et al., 2004, 2005; Spohn et al., 2019). Microbial residues constitute any non-living organic material of microbial origin including necromass and extracellular metabolites (Geyer et al., 2020). Since the transformation of $CO_2$ and release of the fixed carbon via microbial residues vary for different microbial groups (Berg et el., 2011; Miltner et al., 2005), the composition and abundance of microbial communities play a vital role in $CO_2$ fixation rates in soils. High $CO_2$ fixation rates in soils have been reportedly associated with higher abundance of obligate autotrophs and specific bacterial groups like *Proteobacteria* (Long et al., 2015, Xiao et al., 2018). As the microbial communities fixing $CO_2$ are sensitive to changes in edaphic conditions (Berg, 2011; Hügler and Sievert, 2011), various biotic and abiotic predictors of $CO_2$ fixation rates have been identified. Factors like $CO_2$ concentration (Beulig et al., 2016; Spohn et al., 2019; Akinyede et al., 2020), SOC content and quality (Miltner et al., 2005; Šantrůčková et al., 2018; Xiao et al., 2018; Akinyede et al., 2022a), and pH (Santruckova et al., 2005; Long et al., 2015) (Akinyede et al., 2020) affect dark $CO_2$ fixation rates in many soils including those of temperate forests.

Here, we focus on temperature as a factor determining soil $CO_2$ fixation rates. Biological processes are generally faster under higher temperatures due to accelerated rates of enzymatic reactions (Arrhenius, 1889; Van't Hoff, 1898; Davidson and Janssens, 2006). Hence, temperature presumably affects dark $CO_2$ fixation rates, but also the rates of $CO_2$ production through decomposition. If moisture is not limiting, warmer temperatures increase $CO_2$ emission from temperate forest soils (Melillo et al., 2017, 2011, 2002; Walker et al., 2018; Winkler et al., 1996), and the degree of response is similar to depths of 1 meter (Hicks Pries et al., 2017; Soong et al., 2021). Such responses coincided with a reduction in the total SOC pool and were mostly attributed to increased microbial respiration (Melillo et al., 2011). The net change in total $CO_2$ efflux from soil (net soil respiration) includes the effects of temperature on both $CO_2$ production (decomposition) and $CO_2$ fixation. These effects may change with soil depth from the relatively organic carbon-rich surface soils to the more carbon-limited deeper soils.

A previous study describing the influence of temperature on $CO_2$ fixation rates described ~10 times higher fixation rates at 25 °C than at 4 °C in a range of mostly *Alisol* and *Retisol* soils in afro-temperate forest and grassland ecosystems of the lower latitudinal regions (Nel and Cramer, 2019), suggesting potentially large temperature effects on $CO_2$ fixation rates. However, a systematic study comparing the responses of dark $CO_2$

fixation and net soil respiration from temperate forest soils is currently lacking. The relative temperature responses of these processes are important because globally, soil temperatures is projected to warm by ~4 °C until 2100 (based on simulations under the Representative Concentration Pathway (RCP) 8.5 scenario (IPCC, 2013; Soong et al., 2020)).

The temperature sensitivity ($Q_{10}$), the increase of reaction rates for a 10 °C rise in temperature, is a commonly reported value when describing the response of soil microbial processes to higher temperatures (Davidson and Janssens, 2006; Fang et al., 2005; Leifeld and Fuhrer, 2005). Hicks Pries et al. (2017) reported a $Q_{10}$ of 2.4 for net respiration rates in temperate coniferous forest soil. Similar $Q_{10}$ values of between 2 and 3 for net respiration rates have also been described for other soil environments (Conant et al., 2008; Li et al., 2021). Considering that dark $CO_2$ fixation rates and soil respiration rates increase with temperature as described above, and that dark $CO_2$ fixation rates were shown to correlate linearly with net soil respiration rates (Miltner et al., 2005; Šantrůčková et al., 2018), the $Q_{10}$ of dark $CO_2$ fixation rates with depth might correlate with those of net soil respiration rates.

This study describes the temperature sensitivity ($Q_{10}$) of dark $CO_2$ fixation rates and compares it to that of net soil respiration rates across soil profiles of deciduous and coniferous forests, the two temperate forests based on vegetation (Dreiss et al., 2014; Adams et al., 2019). Soils of two acidic forest plots from the Hummelshain forest, Germany dominated by beech (deciduous) and spruce (coniferous) tree species were incubated under two temperature conditions (4 °C and 14 °C). We used a $^{13}CO_2$-labelling approach to quantify dark $CO_2$ fixation rates. We also measured net soil respiration rates and determined the $Q_{10}$ values of both processes across depth. We thus hypothesize that the $Q_{10}$ of dark $CO_2$ fixation rates with depth correlate with those of net soil respiration rates. Using the derived $Q_{10}$ values, we evaluated the potential changes in dark $CO_2$ fixation rates and net soil respiration rates under projected increase in global soil temperature. We further explored the microbial community composition in the beech and spruce soil, with the aim to assess potential differences in the community that might influence dark $CO_2$ fixation rates and thereby, its $Q_{10}$ across temperate forest soil profiles.

## 2 Materials and Methods

### 2.1 Site description and soil classification

The study sites (beech plot: 50°45ʹ28.0ʺ N11°37ʹ21.0ʹE and spruce plot: N50°45ʹ30.0ʺ N 11°37ʹ23.0ʹE) are located within the forested areas of the Hummelshain municipality (~362 m a.s.l.) in Thuringia, central Germany. The study site was established on a former coniferous forest, and it involved the planting of European beech trees within Norway spruce and Scot's pine stands. The main purpose for this conversion was to counteract the low pH of the topsoil under the coniferous stands and thus, biologically activate the forest floors (Graser 1928). The mean annual rainfall in this area is about 630 mm and the mean annual air temperature is around 7.8 °C (Achilles et al., 2021). The two study plots located < 1 km apart are dominated by European beech (*Fagus sylvatica.* Linné) and Norway spruce (*Picea abies* (L.) H. Karst) tree stands, respectively, and feature similar soil geology (Achilles et al., 2020). The soils are mostly sandy (40 – 50 % sand and silt) with a clay enrichment with depth (Table 1; Ad-hoc-AG Boden, 2005; Bormann, 2007) due to the Triassic sandstone bedrock in the Hummelshain area (Achilles et al., 2020). The soils in this region

are predominantly quartz-rich (50 - 60 % quartz), consisting of sandstones and silt-mud stones, and are classified as *Luvisols* with an F-Mull over Loess layer (IUSS Working Group WRB, 2015; Achilles et a., 2020). Both soils feature a low pH (<4), and a high CN ratio (Table 1). The beech soil was slightly lower in SOC, MBC, TN, and moisture content than the spruce soil across depth. The beech soil profile also featured a lower clay but higher sandy texture when compared to the spruce soil profile. Further description of the forest sites and

the soil characteristics in the Hummelshain locality can be found in Achilles et al. (2021).

*Table 1: Geochemical properties of soil cores obtained from beech and spruce soil plots at the Hummelshain forest.* Soil pH, moisture content, soil organic carbon (SOC), carbon/nitrogen (C/N) ratio, microbial biomass carbon (MBC), total nitrogen (TN), MBC/SOC ratio, and natural abundance of $^{13}$C of SOC and MBC, bacterial abundance (16S rRNA gene copies) and soil texture class reported for 3 depths definitions for the beech and spruce soils of the Hummelshain forest. Each reported value represents the mean of three replicate soil cores taken from bulk soils during the sampling campaign.

| Depth (Horizon/ cm) | Beech | | | Spruce | | |
|---|---|---|---|---|---|---|
| | AB (0 - 20) | Bv (20 - 55) | BvT (55 - 100) | AB (0 - 20) | Bv (20 - 55) | BvT (55 - 100) |
| pH | 3.32 ± 0.08 | 3.47 ± 0.03 | 3.13 ± 0.03 | 2.84 ± 0.03 | 3.16 ± 0.02 | 3.07 ± 0.06 |
| Moisture (%) | 8.65 ± 1.30 | 8.28 ± 0.84 | 11.27 ± 1.41 | 10.34 ± 2.22 | 10.95 ± 0.66 | 14.27 ± 1.07 |
| SOC (%) | 0.90 ± 0.12 | 0.28 ± 0.11 | 0.12 ± 0.03 | 1.56 ± 0.07 | 0.33 ± 0.13 | 0.21 ± 0.05 |
| CN ratio | 20.42 ± 1.77 | 11.23 ± 3.02 | 5.59 ± 0.83 | 19.13 ± 1.02 | 9.97 ± 2.37 | 6.38 ± 1.02 |
| MBC (µg C gdw$^{-1}$) | 74.14 ± 3.08 | 21.92 ± 6.53 | 14.26 ± 6.30 | 84.83 ± 9.42 | 25.39 ± 12.85 | 19.43 ± 6.71 |
| TN (%) | 0.04 ± 0.004 | 0.02 ± 0.002 | 0.02 ± 0.002 | 0.08 ± 0.005 | 0.03 ± 0.005 | 0.03 ± 0.002 |
| MBC/SOC (%) | 0.84 ± 0.15 | 0.80 ± 0.09 | 1.05 ± 0.34 | 0.55 ± 0.08 | 0.74 ± 0.10 | 0.96 ± 0.07 |
| δ$^{13}$SOC (‰) | -27.14 ± 0.41 | -25.96 ± 0.16 | -25.37 ± 0.27 | -27.62 ± 0.16 | -25.53 ± 0.31 | -25.25 ± 0.14 |
| δ$^{13}$MBC (‰) | -23.62 ± 0.97 | -22.45 ± 0.59 | -22.71 ± 0.22 | -21.57 ± 0.64 | -22.05 ± 0.85 | -22.67 ± 0.66 |
| 16S rRNA (copies/gdw$^{-1}$) | 1.83 x 10$^9$ ± 8.18 x 10$^8$ | 7.0 x 10$^8$ ± 2.37 x 10$^8$ | 1.19 x 10$^8$ ± 1.06 x 10$^8$ | 2.40 x 10$^9$ ± 3.68 x 10$^8$ | 5.47 x 10$^8$ ± 2.67 x 10$^8$ | 3.89 x 10$^8$ ± 2.22 x 10$^8$ |
| Soil texture class (Ad-hoc-AG Boden, 2005; Bormann, 2007) | Highly silty sand (Su4) | Loamy silty sand (Slu) | Slightly clay loam (Lt2) | Loamy sandy silt (Uls) | Medium clayey silt (Ut3) | Loamy clay (Tl) |

## 2.2 Sampling design

The sampling was carried out in September 2020, towards the end of the summer season. By driving in an 84 mm wide closed auger into the soil with the aid of a motor hammer (Cobra Combi, Atlas Copco AB, Nacka, Sweden), six replicate soil cores, 1 – 2 m apart were obtained from each of the sampling plots, leading to a total of 12 soil cores for the beech and spruce plots. To avoid direct impact from stem flow and to prevent larger

roots from impeding the soil coring process, all soil cores were taken ~2 m away from the base of the trees. Soil sampling began from the mineral horizon while the organic layer was ignored. Three segments were extracted from each soil core by depths chosen according to the similarity of the horizon among all replicate cores to obtain samples representing the AB horizon (0 – 20 cm), Bv horizon (20 – 55 cm), and BvT horizon (55 – 100 cm), for beech plot and the AB horizon (0 – 20 cm), Bv horizon (20 – 55 cm) and BvT horizon (55 – 92 cm) for the spruce sample plot. Soil samples from the same depth intervals of each of the six replicate cores were homogenized in pairs to yield three replicates cores each for the beech and spruce forest plot. Afterwards, all soil samples were sieved using a 2 mm sieve to remove stones and roots prior to the incubation experiments. Fresh subsamples for later DNA extraction and geochemical analysis were also taken, and immediately stored by freezing in liquid nitrogen.

**2.3 Geochemical parameters and isotope measurements**

The total and inorganic carbon and nitrogen concentration, pH, and gravimetric water content as well as carbon isotope signatures of all soil samples were determined as previously described by Akinyede et al. (2020) with values reported in Table 1 and Table S1 (in the supplement). The $^{13}$C signature of the bulk soil total organic carbon was analyzed using an elemental analyzer-isotope ratio mass spectrometer (EA-IRMS) (EA 1110, CE Instruments, Milan, Italy) coupled to a Delta$^+$XL IRMS (Thermo Finnigan, Bremen, Germany) through a ConFlow III interface as previously described (Akinyede et al., 2020). The extraction of microbial biomass carbon content was done by chloroform fumigation extraction (CFE) (Nowak et al., 2015; Vance et al., 1987) using 0.05 M K$_2$SO$_4$ following methods described previously (Akinyede et al., 2020). The microbial biomass carbon content (MBC) extracted as the chloroform soluble carbon content was derived by taking the difference between the dissolved organic carbon (DOC) content in the unfumigated (C$_{unfum}$) and the fumigated soil extract fractions (C$_{fum}$) for all soil samples. Values from all samples were divided by a correction factor $K_{EC}$ (of 0.45) that accounts for the extraction efficiency. This factor corrects for the incomplete release of carbon from the living microbial cells into the solution and is widely applied to different soils (Joergensen and Mueller, 1996; Joergensen et al., 2011; Wu et al., 1990), as CFE only measures the fraction of microbial biomass rendered extractable in K$_2$SO$_4$ solution after lysis with chloroform, which is likely the very labile microbial fraction, (e.g., the cytoplasm) (Ocio and Brookes, 1990; Wu et al., 1990). The MBC content was thus calculated as follows:

$$MBC[mg] = \frac{[C_{fum} - C_{unfum}]}{K_{EC}}$$

(1)

Despite previous studies showing no strong variations in the $K_{EC}$ of 0.45 between soils or incubation temperatures (Martens, 1995; Joergensen et al., 2011), we cannot exclude possible effects resulting from differences in CFE extraction efficiency on our results, especially in comparisons of the rates across the different soil depths or between the beech and spruce soils.

To determine the δ$^{13}$C signature of the bulk soil MBC, the $^{13}$C signature of the DOC from the fumigated and unfumigated CFE fractions was analyzed using an isotope ratio mass spectrometer (DELTA C; stable isotope

monitoring system; Finnigan MAT, Germany) coupled to an elemental analyzer (EA 1100, CE Instruments, Milan, Italy) via a ConFlo III interface as described above. All $^{13}C$ isotope ratios were reported in the delta notation ($\delta$) expressed as $\delta^{13}C$ values ($^{13}C/^{12}C$ ratios) in per mil (‰), relative to the international reference material Vienna Pee Dee Belemnite (V-PDB) (Coplen et al., 2006).

$$\delta^{13}C \; (‰) \; = \left[ \frac{\frac{^{13}C}{^{12}C} sample}{\frac{^{13}C}{^{12}C} reference} - 1 \right] \times 1000$$

(2)

Afterwards, the $\delta^{13}C$ in per mil (‰) of microbial biomass carbon (MBC) was derived by applying an isotope mass balance to the measured $^{13}C$ signals measured for all fumigated and unfumigated DOC fractions from CFE as previously described (Akinyede et al., 2020).

$$\delta^{13}C_{MB} \; [‰] = \frac{\left[ \delta^{13}C_{fum} \times C_{fum} - \delta^{13}C_{unfum} \times C_{unfum} \right]}{C_{fum} - C_{unfum}}$$

(3)

### 2.4 $^{13}C$-CO$_2$ labelling incubation experiment

The CO$_2$ fixation rates were determined using microcosm incubations. Four replicates for each sieved soil sample (60 g wet weight) obtained from all six soil cores in both the beech and spruce sampling plots were placed in sterilized 1000 mL serum bottles, closed with butyl rubber stoppers. The large headspace to soil volume ratio was chosen to ensure minimal changes in the headspace CO$_2$ concentration and the $^{13}C$ isotope signatures, as no further additions to the headspace CO$_2$ were performed throughout the incubation period with labelled $^{13}CO_2$. The four replicate jars were split into two pairs of two replicates each prior to a 4-day preincubation period. The first pair was pre-incubated at 4 °C and the second at 14 °C. Before preincubation, all jars were opened for several minutes to allow the CO$_2$ concentration in the jar to equilibrate with the ambient concentration. After the preincubation period, gas samples were obtained with the aid of a gas syringe for CO$_2$ measurement. Afterwards, the jars were opened, and homogenized soil samples were subsampled for (1) the determination of total/organic carbon and nitrogen content as well as $^{13}C$ isotope signatures of the bulk soil, (2) extraction to determine initial microbial biomass carbon content and its $^{13}C$ isotope signature, and (3) storage for later DNA analysis.

The remainder of the soil (~30 g) was placed in the incubation jar, which was then flushed with synthetic air (75 % N$_2$ and 25 % O$_2$). One replicate of each temperature set was adjusted to a 2 % (v/v) $^{13}C$-CO$_2$ in the headspace and the second replicate was adjusted to 2 % (v/v) headspace $^{12}C$-CO$_2$ concentration, serving as treatments and controls, respectively. All soils exposed to the 2 % (v/v) $^{13}C$-CO$_2$ and controls were then incubated statically in the dark for 21 days under the same temperature as used in the preincubation phase (4 °C and 14 °C). At the end of the incubation period, microcosms were opened and soil samples from all incubations were split into three

parts and geochemical parameters were analyzed as after the pre-incubation phase. Parameters like SOC, MBC, CN ratio, and water content measured after incubation for the beech and spruce plots are described in Table S2 in the supplement and did not differ with temperature and throughout the incubation period. In addition, the $\delta^{13}C$ signals of MBC and SOC from all incubated soil samples were measured as done for the bulk soil prior to the start of the rate measurements.

**2.5 Determination of $CO_2$ fixation rates, respiration rates, and temperature sensitivity ($Q_{10}$).**

To calculate the $CO_2$ fixation rates for all soil incubations at both 4 and 14 °C, the actual $^{13}C/^{12}C$ ratio taken up into the total soil pool and into microbial biomass carbon (MBC) pool was measured as described in section 2.3. This was derived from the measured and derived $^{13}C$ values (for all treatments ($^{13}C$ labelled) and controls ($^{13}C$ unlabelled/natural abundance)) of SOC and MBC respectively.

The $^{13}C/^{12}C$ ratios were calculated based on the $^{13}C/^{12}C$ ratio of the international V-PDB standard as done previously (Akinyede et al., 2020; 2022), where 0.0111802 is taken as the $^{13}C/^{12}C$ ratio of the international V-PDB standard (Werner and Brand, 2001):

$$\frac{^{13}C}{^{12}C} = \left[\frac{\delta^{13}C}{1000} + 1\right] \times 0.0111802$$

(4)

Subsequently, the excess $^{13}C$ ratio for the soil pool and the MBC pool was derived from the increase in $^{13}C/^{12}C$ ratio between the $^{13}C$ labelled treatment and the $^{12}C$ labelled controls ($^{13}C$ natural abundance level) normalized to the respective carbon content of the soil and of the microbial biomass (MBC) as follows:

$$Excess\ ^{13}C[mg] = \frac{^{13}C_{labelled}}{^{12}C_{labelled}} \times MBC/SOC - \frac{^{13}C_{unlabelled}}{^{12}C_{unlabelled}} \times MBC/SOC$$

(5)

These values were then divided by the incubation time and expressed per gram of the bulk soil dry weight and per gram of microbial biomass carbon to obtain the $CO_2$ fixation rates per gram of soil dry weight (g (dw) soil$^{-1}$ d$^{-1}$) and per gram of MBC (g MBC$^{-1}$ d$^{-1}$), respectively.

Following Spohn et al. (2019), the net respiration rates for all soil preincubations were determined based on the difference in the $CO_2$ concentrations of the glass jars measured at the beginning and at the end of the incubation period using a gas chromatograph system for trace gas analysis of air samples (Agilent 6890 GC-FID-ECD-PDD, USA). Gas samples were taken from the headspace of the jars using a gas syringe attached to 250 ml evacuated vials. A period of 30 seconds was allowed for the gas vials to equilibrate with the incubation jars after which the gas vials were disconnected from the vials and connected via a gas line to the gas chromatograph system for $CO_2$ measurement (in ppm). Using the ideal gas equation, net soil respiration rates were calculated according to Dossa et al. (2015), expressed as µg C per gram of soil dry weight per day. As net respiration rates represent $CO_2$ produced minus $CO_2$ fixed, the total $CO_2$ production or decomposition rates were subsequently

derived by adding the net respiration rates to the $CO_2$ fixation rates measured for all beech and spruce soil samples.

The temperature sensitivities of the $CO_2$ fixation (per unit soil and MBC) and net respiration rates (per gram of soil only), as well as the decomposition rate (per gram of soil only), were determined by calculating $Q_{10}$ values according to Leifeld and Fuhrer (2005):

$$Q_{10} = \left(\frac{k_2}{k_1}\right)^{\left(\frac{10}{T_2 - T_1}\right)}$$

(6)

Where $T_2$ and $T_1$ denote the higher and lower temperatures (in °C) at which the soils were incubated, and $k_2$ and $k_1$ respectively represent the corresponding derived $CO_2$ fixation rates/ net respiration/decomposition rates.

**2.6 DNA extraction and 16S rRNA gene sequencing**

DNA was extracted from 0.25 g of all bulk soil and incubation samples using the DNeasy PowerSoil DNA Kit (Qiangen, Hilden, Germany) according to the manufacturer's protocol. For Illumina Miseq sequencing, libraries of amplicon sequences of bacterial 16S rRNA genes were generated. All libraries were prepared with the NEBNext Ultra DNA library prep kit for Illumina (New England Biolabs, Hitchin, UK) using a two-step barcoding approach. For the first step, forward (Bact_341F) and reverse (Bact_785R) primers targeting the V3

to V4 hypervariable regions of the bacterial 16S rRNA gene were used (Klindworth et al., 2013). For Illumina sequencing, the primers were modified with an adaptor overhang which allowed for barcoding in a second PCR step. During the first PCR step, all DNA samples (>10ng/µl) were amplified in a 20 µl reaction volume containing 10 µM of each primer, 0.67 µg/µl of BSA (Bovine Serum Albumin), 5.67 µl nuclease-free water, and 10 µl HotstartTaq Mastermix (Qiagen Hilden, Germany). The PCR conditions used consisted of an initial

denaturation at 95 °C for 45 min, followed by 26 to 30 cycles of denaturation (94 °C for 45 s), annealing (55 °C for 45 s), and extension (72 °C for 45 s), and then a final extension step at 72 °C for 10 min. While samples from the AB and Bv horizon were amplified using 26 to 27 cycles, a few samples from the BvT depth with low DNA concentration were amplified using 30 cycles with the same cycling condition. All amplified sequences from the first PCR step were barcoded in a second PCR step using 1 µl of the initial PCR products, 0.5 µM of

barcoded primer set from Illumina (sequences provided in Table S4 in the supplement), and Ruby Taq Master Mix (Jena Bioscience, Germany) following the cycling conditions of 6 cycles at 95 ∘C for 45 s, 55 ∘C for 45 s and 72 ∘C for 45 s for denaturing, annealing and extension steps respectively. All samples were analyzed by gel electrophoresis using 1 % agarose gel to ensure all amplicons were ~500 bp in length. Subsequently, prepared libraries were sequenced on a Miseq (Illumina, Inc, San Diego, USA) using v3 chemistry (2 x 250 bp).

The raw sequences generated were analyzed using MOTHUR (Schloss et al., 2009; http//: www.mothur.org) and the MOTHUR MiSeq SOP as of 19th January 2021. Paired reads were combined, and sequences were trimmed saving only sequences with the desired length of between 360 bp and 500 bp. Trimmed sequences were aligned to the SILVA reference database v132 release (Quast et al., 2013) and sequences with differences of up to four bases were pre-clustered. Chimeras were removed using Uchime and the GOLD reference database

implemented in MOTHUR (Edgar et al., 2011). Subsequently, the taxonomic classification of the sequences against the SILVA database was performed.

**2.7 Determination of chemolithoautotrophic $CO_2$ fixation potential in the Hummelshain forest soils**

To determine the potential for chemolithoautotrophic $CO_2$ fixation among all soil samples, the abundance of functional genes involved in autotrophic $CO_2$ fixation was first predicted for all bacteria communities. Here, the representative sequences from OTUs generated from MOTHUR were analysed using version 2 (v2.2.0 beta) of the PICRUSt (Phylogenetic Investigation of Communities by Reconstruction of Unobserved States) software package (Douglas et al., 2020). All OTU sequences were de-gaped and placed in a reference taxonomic tree based on the Integrated Microbial Genomes database. EPA-ng and GAPPA tools were used to determine the best position of these placed OTUs in the reference phylogeny (Barbera et al., 2019; Czech and Stamatakis, 2019; www.hmmer.org) after which KEGG orthologs for key enzymes involved in dark $CO_2$ fixation were predicted for each OTU. Using the derived KEGG Orthology (KO) numbers for different key genes for $CO_2$ fixation, the six known autotrophic pathways were deduced for all samples as previously done (Akinyede et al., 2022a, 2020). These include the Calvin Benson Bassham (CBB) pathway (or Calvin cycle), the reductive citric acid (rTCA) pathway, the Wood Ljungdahl (WLP) pathway, the 3-hydroxypropionate/malyl-CoA (3HP) cycle, the 3-hydroxypropionate/4-hydroxybutyrate (HP/HB) cycle, and the dicarboxylate/4-hydroxybutyrate (DC/HB) cycle.

Based on PICRUSt2 predictions, the abundance of functional genes belonging to two $CO_2$ fixation pathways: the Calvin cycle and the rTCA pathway, were determined by quantitative PCR for the bulk soil as well in all soil samples incubated at 4 and 14 °C. Gene abundance of bacterial *16S rRNA*, RuBisCO marker genes (*cbbL IA*, *cbbl IC*, *cbbM*) for the Calvin Benson Basham cycle and ATP citrate lyase genes, *aclA* belonging to the reductive citric acid cycle was determined by quantitative PCR on the CFX 96 Touch real time PCR system (Bio rad, Singapore) using Maxima SYBR Green Master mix (Agilent, CA, U.S.A). Primer pair Bac *8Fmod*/*Bac 338R* was used to target the *16S rRNA* genes ((Loy et al., 2002; Daims et al., 1999) while *F-cbbL IA/R-cbbL IA*, *F-cbbL IC/R-cbbL IC,* and *F-cbbM/R-cbbM* were used to target both the form I (*cbbL IA* and *cbbl IC*) and form II (*cbbM*) RuBisCO marker genes (Alfreider et al., 2012, 2003) which is specific for both obligate and facultative chemolithoautotrophic bacteria groups like *Proteobacteria* (Selesi et al., 2005). Primer pair *F-g-acl-Nit/R-g-acl-Nit* was used to target the alpha subunit of ATP citrate lyase (*aclA*) gene, which is specific for nitrite-oxidizing bacteria and complete ammonia-oxidizing (comammox) bacteria e.g., Nitrospira (Alfreider et al., 2018). All cycling conditions and standards used for quantification are found in Akinyede et al. (2020) and Herrmann et al. (2012, 2015). Due to the absence of reliable standardized qPCR protocol and primer sets to target genes for the WLP pathway and the rest of the other autotrophic pathways, the presence of these pathways was based only on the predictions by PICRUSt2.

**2.8 Statistical analysis**

We compared the $CO_2$ fixation rates per gram of soil and per gram of MBC between all soil samples incubated at 4 and 14 °C using Student's t-test. To compare the respective $Q_{10}$ values between the beech and spruce profile and across individual horizons, ANCOVA and one-way ANOVA with Tukey's test were conducted,

respectively. To compare other parameters between the beech and spruce soil profiles, e.g., net soil respiration rates between 4 and 14 °C, and $^{13}$C signal of SOC and MBC, ANCOVA was also conducted. When comparing parameters between the beech and spruce soils using ANCOVA, soil depth also accounted for the variability in the measured parameters and was used as the covariate in the analysis. When deriving the $CO_2$ fixation and net soil respiration rates under projected future temperatures increase, (from 8 °C to 12 °C), the mean $Q_{10}$ values for the beech and spruce profiles were used in the $Q_{10}$ equation described in Eq. (6) of the method sect 2.5. As rates at 4 °C were low, and in some samples, below the detection limit (e.g., net respiration rates at the beech soil BvT depth), the rates per gram of soil measured at 14 °C were used (as "$k_2$, $T_2$") in the $Q_{10}$ equation to derive rates at 8 °C ("$k_1$, $T_1$"). The derived rates at 8 °C were then used (as "$k_1$, $T_1$") for the subsequent derivation of rates at 12 °C ("$k_2$, $T_2$"). Following Geyer et al., (2020), we quantified the proportion of excess $^{13}$C transferred into the SOC pool from the MBC pool via microbial residues as the total amount of excess $^{13}$C fixed in the SOC pool minus the excess $^{13}$C fixed in the intact MBC pool.

The variations in the bacterial community between the beech and spruce soil and with temperature were determined by measuring the beta diversity. Beta diversity was measured by performing Principal Coordinate Analysis (PCoA) based on Bray-Curtis dissimilarity using the package, vegan contained in R (Oksanen et al., 2008). Here, the bacterial communities from all beech and spruce soil samples were clustered based on their similarity/dissimilarity between the soils and with temperature. To determine the significance of the factors accounting for OTU variances shown in the PCoA plot between the two soils, across individual soil depth, and between temperatures, Permutational Multivariant Analysis of Variance (PERMANOVA) was performed with 999 permutations using "adonis" functions. Differences in the abundance of all predicted and quantified $CO_2$ fixation genes between the beech and spruce soil were analysed using ANOVA and Tukey's test. For all statistical tests, differences with $p < 0.05$ were considered statistically significant. All statistical analyses were conducted with the R environment (v.3.6.1) and RStudio (v1.1.463).

## 3. Results

### 3.1 Effects of temperature on dark $CO_2$ fixation rates in beech and spruce soils.

All soil incubations exposed to $^{13}CO_2$ were significantly enriched in $\delta^{13}$C relative to the controls both at 4 °C and 14 °C, indicating dark $CO_2$ fixation (Fig. S1 in the supplement). In the top depths of both the beech and spruce soils, significantly higher $CO_2$ fixation rates were observed at 14 °C than at 4 °C. For the top AB horizon of the beech soil, $CO_2$ fixation rates expressed in relation to soil dry weight (µg C g (dw) soil$^{-1}$ d$^{-1}$) (Fig. 1A) were almost 2 times higher with $0.033 \pm 0.006$ µg C g (dw) soil$^{-1}$ d$^{-1}$ at 14 °C compared to $0.018 \pm 0.006$ µg C g (dw) soil$^{-1}$ d$^{-1}$ at 4 °C ($p = 0.04$; Student's t.test). Similarly, the top AB depth of the spruce soil, also featured ~2 times higher fixation rates at 14 °C with $0.030 \pm 0.003$ µg C g (dw) soil$^{-1}$ d$^{-1}$ than at 4 °C with $0.014 \pm 0.002$ µg C g (dw) soil$^{-1}$ d$^{-1}$ ($p = 0.005$). In the lower depths, however, no significant differences in fixation rates expressed per gram soil dry weight were observed between soils incubated at 4 °C and 14 °C for either the beech ($p = 0.3$ and $p = 0.6$ at the Bv and BvT horizons, respectively) or spruce soils ($p = 0.2$ and $p = 0.4$ at the Bv and BvT horizons, respectively). While we observed an expected decrease in fixation rates per gram of soil with depth in both soils due to the decreasing SOC content, there were no significant differences in rates between the beech and spruce soil, neither at 4 °C nor at 14 °C. Across the depth profiles, changes in rates with

temperature did not differ between the spruce (1.5 – 3.2-fold changes) and the beech soil (0.9 – 2.7-fold changes) ($p = 0.08$) as both soils showed a 70 - 90% increase in $^{13}C$ signal with temperature (Figure S1).

When expressed in relation to microbial biomass carbon (MBC), dark $CO_2$ fixation rates in the top AB horizon of the spruce soil were 1.6 times higher at 14 °C with 145.95 ± 27.13 µg C g MBC$^{-1}$ d$^{-1}$ than at 4 °C with 88.29 ± 17.12 µg C g MBC$^{-1}$ d$^{-1}$ ($p = 0.04$) (Fig. 1B). For the beech soil, however, values in the top AB depth were

similar at 4 °C and 14 °C ($p = 0.3$) with 108.18 ± 8.82 µg C g MBC$^{-1}$ d$^{-1}$ and 125.11 ± 23.76 µg C g MBC$^{-1}$ d$^{-1}$, respectively. In the lower horizons, no significant differences between temperature treatments were observed for the two soils. Rates expressed per gram MBC was approximately constant with depth, excepting the BvT horizon of the beech soil that had lower rates. Taken together across depth profiles, stronger differences with temperature were observed for the spruce (1.3 – 2.6-fold changes) than the beech soil (0.9 - 1.4-fold changes) ($p$

$= 0.003$). These differences in rates reflect the observed difference in the $^{13}C$ signals of MBC with temperature between the soils. While up to 124% increase in $^{13}C$ signal of MBC was found for the spruce soil, the beech soil showed no more than 23% increase in $^{13}C$ signal of MBC with temperature (Figure S1). These differences between the beech and spruce soil suggest that drivers of dark $CO_2$ fixation may differ between soils.

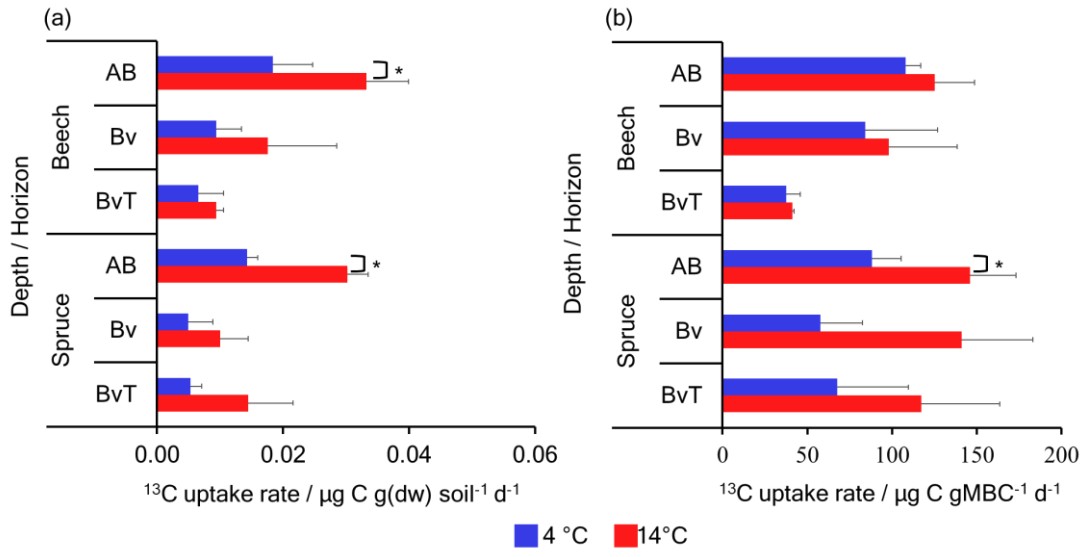

**Figure 1: Dark CO$_2$ fixation rate measured from soil microcosms supplemented with 2 % $^{13}$CO$_2$ at 4 and 14 °C.** Shown are (a) $^{13}C$ uptake rates in soil expressed in µg C g(dw) soil$^{-1}$ d$^{-1}$ (µg carbon per gram dry weight (dw) of soil per day) and (b) $^{13}C$ uptake rates into MBC expressed in µg C gMBC$^{-1}$ d$^{-1}$ (µg carbon per gram microbial biomass carbon per day) after 21 days of incubation with 2 % $^{13}$CO$_2$ at 4 (blue bars) and 14°C (red bars) across three horizons of the beech and spruce soils. Error bars indicate the standard deviation of incubations from three replicate soil cores. * denotes $p < 0.05$.

**3.2 Q$_{10}$ of dark CO$_2$ fixation rates for beech and spruce soil profiles.**

The Q$_{10}$ values, the factor by which CO$_2$ fixation rates differed with the 10 °C rise in temperature, were 1.81 ± 0.17 across depths (for rates per gram of soil) for the beech soil, and 2.34 ± 0.21 for the spruce soil (Fig. 2A) with a mean Q$_{10}$ value of 2.07 ± 0.34 for all beech and spruce soils. Both the beech and spruce soils showed large variability in the Q$_{10}$ values with depth with values ranging from 1.97 ± 0.84 at the AB depth and 1.63 ±

0.86 at the bottom BvT depth of the beech soil and from 2.11 ± 0.07 to 2.53 ± 0.70 through the spruce soil profile. Thus, no significant differences between corresponding depths of both soil profiles were observed. ($p = 0.81$, $p = 0.32$ and $p = 0.23$ for the AB, BV and BVT horizons respectively). However, differing trends across

individual depths for the beech and spruce soil were observed, with $Q_{10}$ values decreasing with depth in beech soil, but increasing with depth in spruce soil ($R^2 = 0.92$, $p = 3.2 \times 10^{-8}$, ANCOVA).

We also calculated the $Q_{10}$ for rates per gram of MBC, as the microbial cells are responsible for dark $CO_2$ fixation and should be primarily affected by temperature. Compared to the $Q_{10}$ based on soil dry weight, the $Q_{10}$ based on MBC was lower in the beech soil ($p = 0.008$) which be linked to the smaller differences in the $^{13}C$ signal of MBC and calculated $CO_2$ fixation rates with temperature. For the spruce soil, no difference between the $Q_{10}$ based on MBC and the $Q_{10}$ based on soil dry weight was observed ($p = 0.13$). As a result, the spruce soil

profile featured a higher mean $Q_{10}$ based on rates per gram of MBC with $1.9 \pm 0.63$ (Fig. 2B) than the beech soil profile with $1.1 \pm 0.20$ across depth ($p = 0.003$; ANOVA and Tukey's test).

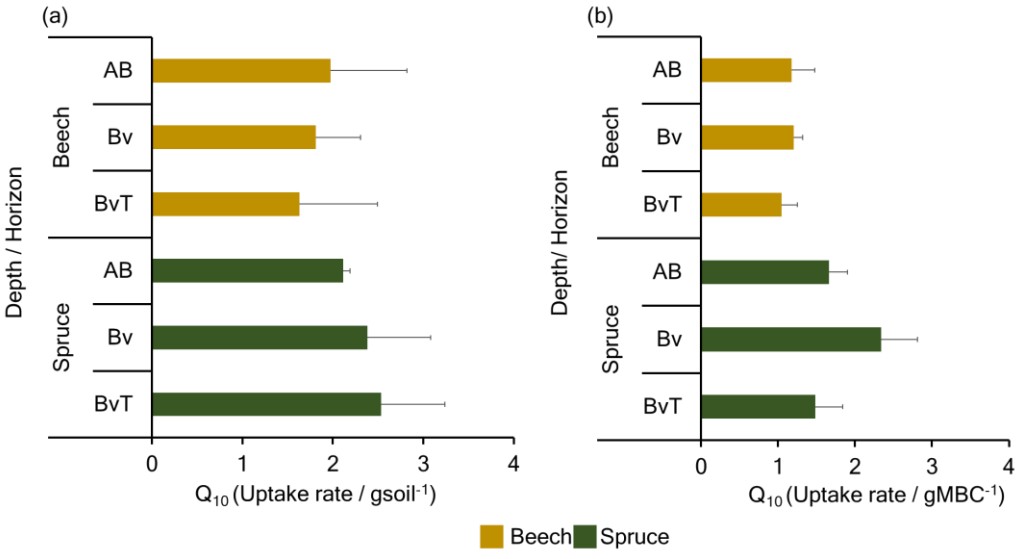

**Figure 2: Temperature sensitivity ($Q_{10}$) of dark $CO_2$ fixation rate measured from soil microcosms supplemented with 2 % $^{13}CO_2$ at 4 and 14 °C.** Shown are the $Q_{10}$ (temperature sensitivity) values of dark $CO_2$ fixation rates derived from
fixation rates expressed in (a) µg C g(dw) soil$^{-1}$ d$^{-1}$ (µg carbon per gram dry weight (dw) of soil per day) and in (b) µg C gMBC$^{-1}$ d$^{-1}$ (µg carbon per gram microbial biomass carbon per day) after 21 days of incubation with 2 % $^{13}CO_2$ across three horizons in the beech and spruce soils.

### 3.3 Allocation of $^{13}C$ in the beech and spruce soils.

As both the fixation rates and $Q_{10}$ values differed between the beech and spruce soil, we aimed to determine if
this was reflected by differences in the partitioning or transfer of the fixed $^{13}C$ via microbial residues between the MBC and SOC pools. On average, the $^{13}C$ signals of the MBC pool were significantly lower across the beech soil profile compared to the spruce soil profile at 14 °C ($R^2 = 0.91$, $p = 5.24 \times 10^{-8}$, ANCOVA) with no clear difference observed between forest types at 4 °C (Fig. S2 in the supplement). In contrast, the $^{13}C$ signatures measured in SOC were on average higher in the beech than in the spruce soil across depth for soils incubated
both at 4 °C and at 14 °C ($R^2 = 0.98$, $p = 3.90 \times 10^{-12}$ for 4 °C and $R^2 = 0.96$, $p = 1.05 \times 10^{-10}$ for 14 °C, ANCOVA). As a result, a higher proportion of fixed $^{13}C$ was found to be allocated to the MBC pool in the spruce soil with up to 64 % compared to the beech soil with up to 32 % (Fig. 3). Hence, in the beech soil, a greater amount of $^{13}C$ allocation into the SOC pool was observed compared to the spruce soil. In general, higher temperatures were associated with larger increase in $^{13}C$ allocation to SOC compared to MBC pools in the beech

and spruce soils (Fig. 3). For instance, at the AB depth of the beech soil, $^{13}$C fixed into SOC increased from 67 % to 81 % and in the spruce soil, it went up from 36 % to 63 %.

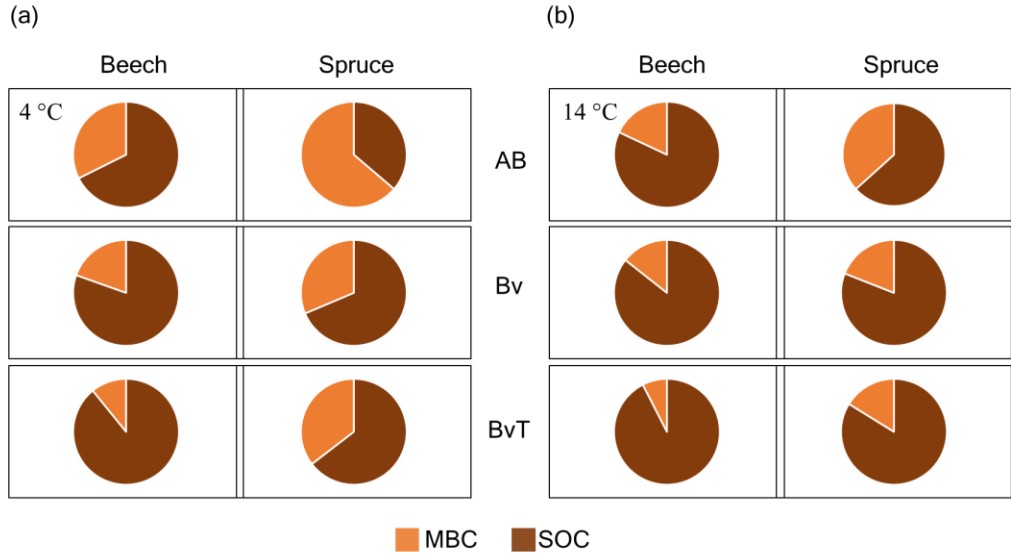

**Figure 3: Proportion of fixed carbon recovered in MBC and SOC pools of the beech and spruce soils.** The pie chart shows the relative proportions of microbially derived $^{13}CO_2$ into MBC (orange fractions) and SOC (brown fractions) pools
expressed as a percentage of the total fixed carbon after 21 days of incubation with 2 % $^{13}CO_2$ for soils incubated at (a) 4 and (b) 14°C across three horizons (AB, Bv, BvT) in beech and spruce soils. Values are the mean of 3 replicate incubations.

### 3.4 Effects of temperature on net soil respiration rates.

In addition to the $CO_2$ fixation rates, we also determined net soil respiration rates during the pre-incubation phase. As expected, net respiration rates across all beech and spruce soil samples were 20 -70 times higher than
$CO_2$ fixation rates ($p = 0.005$) with values as high as 2.89 ± 1.26 µg C g (dw) soil$^{-1}$ d$^{-1}$ and 2.31 ± 0.9 µg C g (dw) soil$^{-1}$ d$^{-1}$ at 14 °C at the top AB horizon of the beech and spruce soils, respectively. Net respiration rates were higher in all soils incubated under 14 °C than at 4 °C for both the beech ($R^2 = 0.73$, $p = 0.03$, ANCOVA) and spruce ($R^2 = 0.58$, $p = 0.02$, ANCOVA) profiles (Fig. 4). As rates were highly variable across replicates, no significant differences between the beech and spruce soil or with depth were observed.

In response to warming, the $Q_{10}$ of net respiration rates per gram of soil for the beech and spruce soils were 2.87 ± 0.81 and 3.06 ± 0.78, respectively. Taken together, the mean $Q_{10}$ for net respiration rates across the beech and spruce soil profiles at 2.98 ± 0.69 was significantly higher than the $Q_{10}$ of fixation rates relative to soil dry weight ($R^2 = 0.95$, $p = 3.0$ x $10^{-5}$, ANCOVA). Values ranged between 2.29 ± 0.004 and 3.44 ± 1.43 across the two AB and Bv depths in the beech soil and between 2.60 ± 0.12 to 3.96 ± 3.38 across all three depths of the
spruce soil. Due to the high variations in net respiration rates among the soil samples, the $Q_{10}$ values did not differ significantly between the beech and spruce soil and across the individual depth profiles. As net respiration rates were much higher than $CO_2$ fixation rates, the derived decomposition rates were nearly the same as the net respiration rates for the beech and spruce soils. Hence the $Q_{10}$ values were also similar having a mean value of 2.95 ± 1.34 (Table S3 in the supplement).

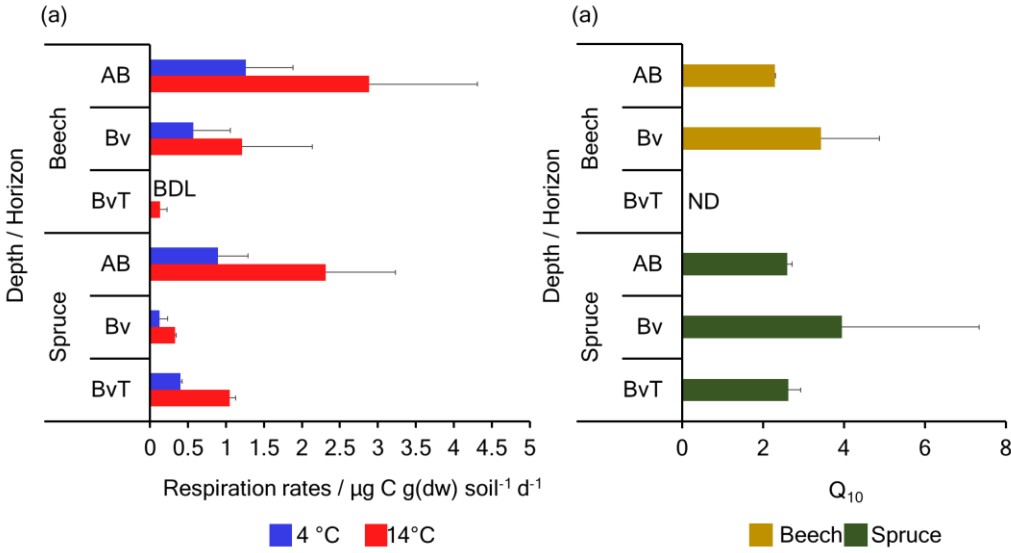


**Figure 4: Net respiration rates and Q$_{10}$ values measured from soil microcosms incubated at 4 and 14 °C.** Shown are (a) net respiration rates in beech and spruce soils expressed in µg C g(dw) soil$^{-1}$ d$^{-1}$ (µg carbon per gram dry weight (dw) of soil per day) at 4 (blue bars) and 14°C (red bars) and (b) Q$_{10}$ (temperature sensitivity) of net respiration rates measured after 4 days of pre-incubation in beech (yellow bars) and spruce soils (green bars) across depth. Error bars indicate the standard deviation of incubations from three replicate soil cores. BDL and ND denote values "below detection limit" and values "not detected" respectively.


### 3.5 Bacterial communities of the Hummelshain forest soils.

The lower $^{13}$C allocation in MBC but higher allocations to SOC for beech than for spruce soils indicated a higher turnover of fixed $^{13}$C from MBC to SOC in the beech soil compared to the spruce soil. We thus further

checked if this higher turnover was accompanied by differences in the overall bacterial community composition and abundance. We investigated the 16S rRNA gene amplicons at operational taxonomic unit (OTU) level and determined 16S rRNA gene copies by qPCR (Table 1 and Fig. S3 in the supplement). Principal coordinate analysis (PCoA) revealed differences in the composition between the beech and spruce soil ($R^2$ = 0.23, p = 0.001, PERMANOVA, Fig. 5A). This was most pronounced in the top AB horizon ($R^2$ = 0.67, p = 0.003).

Furthermore, the bacterial community composition also differed with soil depth ($R^2$ = 0.30, p = 0.001). As expected, the microbial abundance decreased with depth in both soils (Table 1). However, no difference in the microbial abundance between soils was observed at comparable depths. With respect to temperature, no shifts in the community composition were found in either beech ($R^2$ = 0.014, $p$ = 0.99) or spruce ($R^2$ = 0.015, $p$ = 0.98) soils (Fig. 5B). Likewise, the microbial abundances did not differ with temperature (Table S2 in the

supplement).

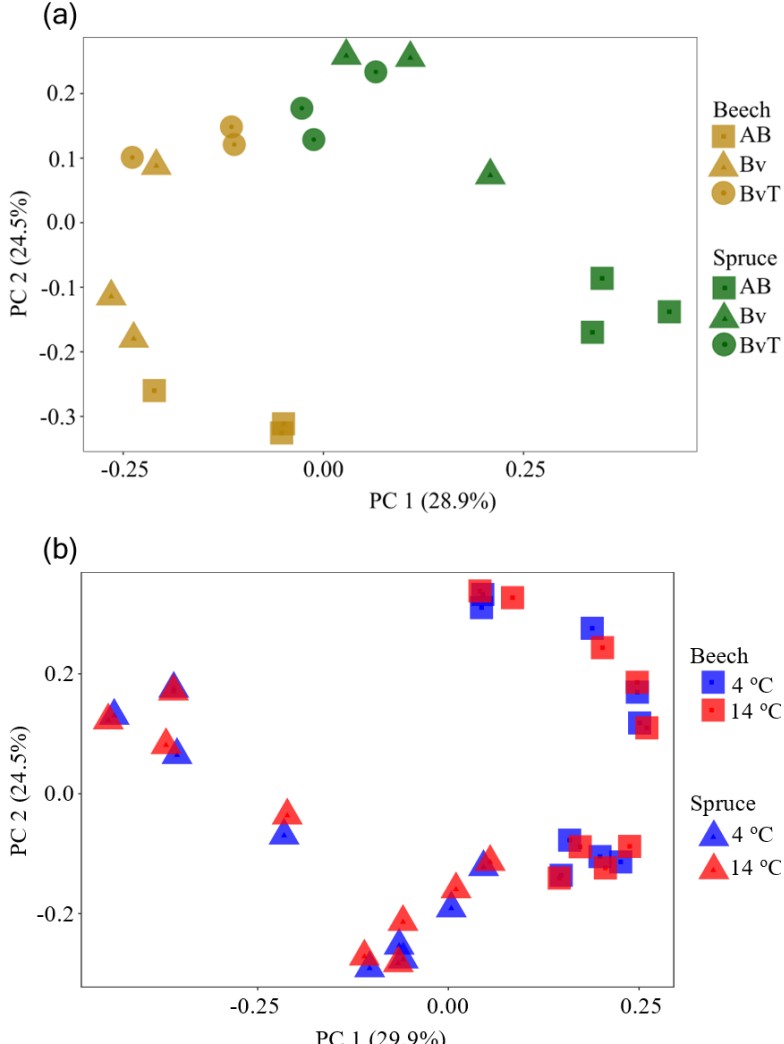

**Figure 5: Bacterial community composition and community structure from beech and spruce bulk soil.** Shown are the bacterial community structure of (a) the beech (yellow symbols) and spruce (green symbols) bulk soils before incubation and of (b) beech and spruce soils incubated with 2 % $^{13}CO_2$ at 4 (blue symbols) and 14°C (red symbols). PCoA plot is based on OTU level analysis (Bray-Curtis dissimilarity) of 16S rRNA gene amplicons generated by Illumina Miseq sequencing with three independent data points per depth obtained from beech and spruce soils.

### 3.6 Abundance of genes for $CO_2$ fixation.

Based on presumed differences in residue formation and in the community composition between the soils, we speculate that the potential key players in the beech soil were composed of a higher proportion of groups with faster life cycles when compared to the spruce soil. As the rich SOC content in forest soils generally promotes faster growth of heterotrophs over chemolithoautotrophs, we further hypothesised that the beech bulk soil contains lower fractions of autotrophs compared to the spruce soil. We used PICRUSt2 to predict and quantify the genetic potential for $CO_2$ fixation in both soils to test this hypothesis. Predicted autotrophic OTUs made up ~11 % of the total bacterial community in all samples. Most of the autotrophic OTUs were predicted to possess genes affiliated with RuBisCO of the CBB pathway for $CO_2$ fixation, with ~9 %, while genes of the WLP and the rTCA pathways were predicted in ~2 % and 0.1 % of the OTUs, respectively. The spruce bulk soil featured higher abundances of OTUs predicted to possess the RuBisCO gene than the beech bulk soil (Fig. 6A), with significantly higher proportions in the AB horizon ($p = 0.007$).

Quantitative PCR of marker genes coding for the CBB (RuBisCO (*cbbL* IA, *cbbL* IC, *cbbM*) and the rTCA pathway (ATP-citrate lyase alpha subunit (*aclA*)) was done to confirm the predicted potential for autotrophic $CO_2$ fixation. Of the detected gene variants, the *cbbL* IC gene was the most abundant with up to 5 % of the bacterial 16S rRNA gene copies in both soils, whereas other RuBisCO and *aclA* genes constituted less than 1 % (Fig. S4 in the supplement). The proportions of *cbbL* IC genes across the depths were significantly higher in the spruce than in the beech soil at 4 and 14°C ($p = 0.007$, $p = 1.03 \times 10^{-6}$, respectively) (Fig. 6B). In the bulk soil, however, this was only observed in the BvT horizon ($p = 0.03$).

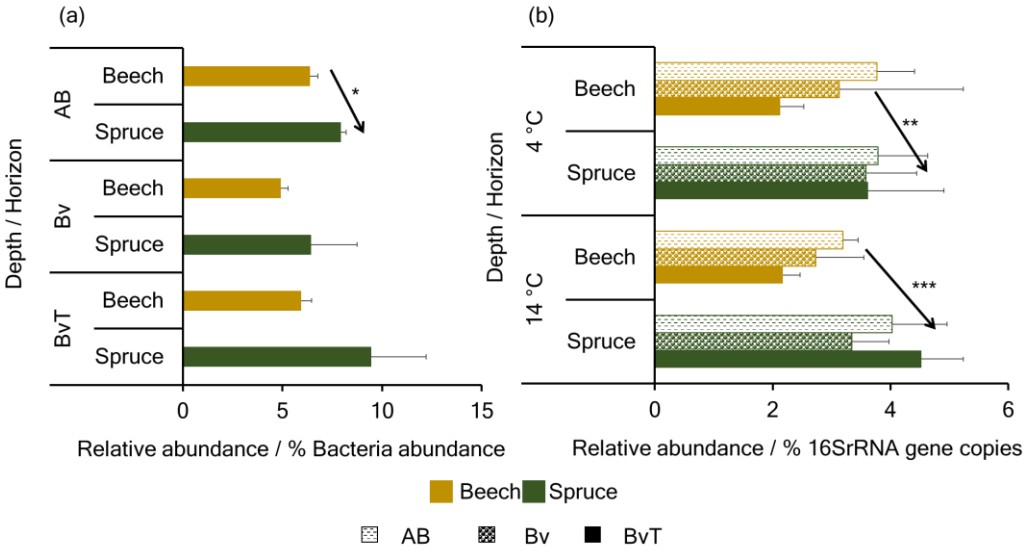

**Figure 6: Relative abundance of RuBisCO genes in the beech and spruce soil.** Shown are the relative abundances of (a) predicted and (b) quantified RuBisCO genes coding for the CBB pathway in the beech and spruce soil. Data in (a) is based on predictions by PICRUSt2 analysis of bacterial 16S rRNA gene amplicon sequence data for the beech and spruce bulk soil while data in (b) is acquired by qPCR of cbbL IC genes for both soils incubated at 4 °C and 14 °C. *, **, *** denote $p < 0.05$, 0.01 and 0.001, respectively.

**4 Discussion**

This study shows that the derived $Q_{10}$ of dark $CO_2$ fixation rates per gram of soil, with a mean value of 2.07 across the beech and spruce soil depths, were significantly lower than the average $Q_{10}$ of net soil respiration rates per gram of soil with 2.98 for both soils, which suggests that soil respiration is more sensitive to warming than $CO_2$ fixation. Our $Q_{10}$ values of net soil respiration rates fall in the range of those reported for agricultural soils (1.5) and temperate mixed forest soils (3.1) (Fang et al., 2005; Conant et al., 2008; Hicks Pries et al., 2017; Li et al., 2021). For $Q_{10}$ values of dark $CO_2$ fixation rates, a similar value of ~2.5 was extrapolated for afro-temperate forest soils (Nel and Cramer, 2019).

Comparing the responses of dark $CO_2$ fixation and net soil respiration to temperature for the same soil is important in understanding the dynamics of SOC fluxes within the context of global climate change. Since dark $CO_2$ fixation can recycle up to 4 % of $CO_2$ respired from temperate forest soils (Spohn et al., 2019), higher $CO_2$ fixation rates might as well be affecting the magnitude of SOC losses from temperate forest soils under warming. By assuming a $Q_{10}$ of 2.07 and 2.98 for dark $CO_2$ fixation rate and net soil respiration rates, respectively, we extrapolated the effect of future warming on the forest SOC fluxes. With a 4 °C increase in

mean annual temperate forest soil temperature (~8 °C now to 12 °C by 2100) to 1 m deep by the end of this century, (IPCC, 2013; Soong et al., 2020), dark $CO_2$ fixation rates to 1 m depth would increase by 33 % while net soil respiration rates would increase by 55 %. This indicates that future increase in net soil respiration might be 1.16 times higher than $CO_2$ fixation upon 4 °C warming. Hence, the potential for dark $CO_2$ fixation to recycle or modulate carbon respired from temperate forest soils could decrease under future warming scenarios. However, the temperature response of dark $CO_2$ fixation and respiration in soils is likely also affected by varying temperatures occurring in different temperate forest biomes. Differences in carbon allocation between MBC and SOC also show that not all components of the soil carbon cycle will have the same response to soil temperature changes. Furthermore, higher temperature might alter primary production and root exudation resulting in changes in soil carbon inputs and consequently, soil pore space $CO_2$ concentrations and effluxes (Jakoby et al., 2020; Way and Oren, 2010; Yin et al., 2013). Thus, the estimates presented here are associated with a range of uncertainty.

Our measured soil $CO_2$ production in all soil incubations does not represent decomposition rates but the net soil respiration rates, as these rates include the effects of temperature on both $CO_2$ production (decomposition) and $CO_2$ fixation, both processes occurring simultaneously (Braun et al., 2021). Thus, to accurately derive the decomposition rates, $CO_2$ fixation rates have to be added to measured net soil respiration rates. Although our measured $CO_2$ fixation rates were very small with only marginal effects on the $Q_{10}$ of soil $CO_2$ production, that is, a $Q_{10}$ of 2.98 vs 2.95 for net soil respiration and decomposition rate respectively. Nevertheless, dark $CO_2$ fixation may result in an overestimation of $Q_{10}$ of decomposition rates if only net soil respiration is measured. This is especially the case in scenarios where high $CO_2$ fixation rates are expected.

Unexpectedly, the two soils, beech and spruce, showed differences in their temperature response. Although $Q_{10}$ values for $CO_2$ fixation rates per gram of soil were similar between the beech and spruce topsoil, the $Q_{10}$ differed in their depth trends, with decreasing $Q_{10}$ with depth in the beech soil and increasing $Q_{10}$ with depth in the spruce soil. Furthermore, while both soils showed similar temperature responses in terms of rates of $CO_2$ production per gram of soil, the temperature effect of $CO_2$ fixation rates expressed per gram of MBC were smaller in the beech than in the spruce soil. The lower $Q_{10}$ in the beech soil was accompanied by higher proportions of newly fixed $^{13}C$ in the SOC pool but lower proportions in the MBC pool when compared to the spruce soil, especially for soils incubated at 14 °C. This suggested that there was a higher transfer of microbially derived carbon from the MBC pool into the SOC pool in the beech soil. Through microbial residue formation, fixed $^{13}C$ is transferred from the MBC into the SOC pool (Geyer et al., 2020; Miltner et al., 2012). A higher rate of residue formation in the beech soil will lead to a lower fraction of fixed $^{13}C$ remaining in the MBC pool and thereby, an underestimation of the $CO_2$ fixation rates per gram of MBC when compared to the spruce soil. Hence, a higher rate of microbial residue transfer in our 21-day incubations might explain the lower $Q_{10}$ of fixation rates per gram of MBC in the beech than in the spruce soil. Such rapid residue formation is not uncommon in soils, as it was observed in as little as six hours in a temperate forest soil (Geyer et al. (2020). Microbial biomass can also turnover as necromass within a few days to weeks in soils (Kästner et al., 2021; Miltner et al., 2012) with turnover times of 18 – 21 days reported in agricultural soils (Cheng, 2009), and 33 days in temperate forest soils (Spohn et al., 2016). Thus, the formation of microbial residues can be observed within the timescale of our incubation experiment.

Accelerated microbial residue formation in the beech compared to the spruce soil might have been related to differences in soil abiotic parameters, in particular, factors differentially affecting either the lifespan (turnover) of microbial cells or the formation of extracellular metabolites from living cells. The biggest difference between the soils was soil texture, with lower clay and higher sand content in the beech compared to the spruce soil. Soil texture is known to affect microbial biomass turnover in soils (Van Veen et al., 1984; Sakamoto and Hodono,

2000; Prévost-Bouré et al., 2014), with high clay content soils often associated with slow biomass turnover into necromass compared to low clay content soils (Ali et al., 2020; Gregorich et al., 1991; Van Veen et al., 1985), due to the capacity of clay-rich soils to protect or preserve microbial cells, reducing overall death rate (Van Veen et al., 1985, 1984). Interaction of microbial biomass with the negatively charged clay mineral particles was suggested as the mechanism causing biomass stability (Ali et al., 2020; Six et al., 2006). Clay-microbe

interactions may promote microbial growth by maintaining optimal pH range (Stotzky et al., 1966) and helping to adsorb metabolites inhibitory to microbial growth (Martin et al., 1976). The fine particles and small pore space characteristic of clay-rich soils also lead to a higher water holding capacity (Fan et al., 2004; Jommi and Della Vecchia, 2016; Miltner et al., 2009; Tsubo et al., 2007), resulting in the observed higher moisture content in the spruce compared to the beech soil. This higher moisture typical for clay-rich soils might partly be

responsible for protecting microbes against moisture limitations when compared to sandy soils (Meisner et al., 2018; Schnürer et al., 1986; Bitton et al., 1976). Furthermore, small pores also restrict the access of higher organisms like protozoa, providing protection against predation (Elliott et al., 1980; Rutherford and Juma, 1992). Microbial protection promotes recycling or transfer of microbial extracellular products among the living communities, thus, preventing further release into the soil pool (Gregorich et al., 1991). All of these mechanisms

imply that the formation of microbial residues in the more clay-rich spruce soils should be slower than in the beech soil, as we observed. Due to the presence of larger mineral surface areas of clay in the spruce soils, association with clay surfaces can also lower residue formation and the amount of $^{13}$C label transferred would be a smaller proportion of total SOC. However, as the mineral composition of the soils was not measured, we cannot verify this assumption.

The higher microbial residue formation in the beech soil was accompanied by a different community composition and a lower proportion of genes for chemolithoautotrophic $CO_2$ fixation. Considering that both soils featured a similar abundance of the total bacterial community, this implied a higher proportion of heterotrophs among the beech soil communities. Growth of heterotrophs is favoured by high amounts of simple and complex carbon substrates released as root exudates (Huang et al., 2022; Li et al., 2018; Lladó et al., 2017;

Vijay et al., 2019). Whereas soils are usually deficient in reduced inorganic compounds (Jones et al., 2018) required as energy sources for autotrophic growth (Brock et al., 2003; Berg, 2011). The redox potential of half reactions utilized by chemolithoautotrophs for energy often leads to low energy yield than commonly observed for heterotrophs, causing chemolithoautotrophs to grow slower (Hooper and DiSpirito, 2013; Madigan et al., 2015; In 't Zandt et al., 2018). As cell growth correlates with microbial residue formation (Geyer et al., 2020;

Hagerty et al., 2014; Kästner et al., 2021), it is likely that heterotrophs in soils also form residues at faster rates than their chemolithoautotrophic counterparts. Hence, the suggested higher proportion of heterotrophs in the beech soil could also explain the higher rate of microbial residue formation observed when compared to the spruce soil. In both the beech and spruce soil, majority of the chemolithoautotrophic genes were affiliated to facultative autotrophs or mixotrophs which can also utilize SOC as a carbon source for growth (Yuan et al.,

2012). This versatility allows them to be more active or grow faster than obligate autotrophs (Madigan et al., 2015). Hence, mixotrophs might contribute to microbial residue formation into the SOC pool especially in the beech soils.

The proportion of labelled carbon transferred to the SOC pool increased with temperatures in both beech and spruce soils and across all depths, indicating higher inputs of microbial residues under warming. Higher temperatures have been often reported to increase inputs of microbial residues into soil (Ding et al., 2019; Hagerty et al., 2014; Li et al., 2019). Increasing soil temperature by 10 °C (15 to 25 °C) was reported to double the specific death rate of microbial communities in soil due to increased protein turnover (Joergensen et al., 1990). Increased microbial residue formation of soil microbial biomass was suggested to result from higher rates of enzymatic activities or changes in the abundance and composition of the soil microbial community (Ding et al., 2019; Hagerty et al., 2014). However, we did not find changes in the composition and abundance of the microbial community with warming in both the beech and spruce soils during this short incubation time. Previous studies have shown that even after four or five years of warming, no increase in bacterial and fungal biomass was observed for a temperate forest soil (Schindlbacher et al., 2011) and it can take up to a decade to detect temperature related changes in the soil community composition (Rinnan et al., 2009, 2007). In agreement, DeAngelis et al. (2015) revealed that 5 °C soil warming had a significant impact on bacterial community structure in mixed deciduous temperate forest soils only after 20 years of warming. Authors suggest that soil communities are more likely driven by gradual warming-induced changes in aboveground plant biomass and composition, and associated shifts in carbon substrate, moisture, and nutrient conditions rather than just elevated soil temperature effects (Sarathchandra et al., 1989; Rinnan et al., 2007; Frey et al., 2008; DeAngelis et al., 2015). This indicates that increases in dark $CO_2$ fixation in temperate forest soils as a response to short-term warming may not be caused by an increased microbial abundance or a shift in community composition, but likely by an increase in the formation and release of microbial residues.

**5 Conclusion**

In response to warming, we measured an average $Q_{10}$ of 2.07 for $CO_2$ fixation rates per gram of soil across 1 m depth profiles for soils dominated by deciduous-beech and coniferous-spruce trees. As net soil respiration rates across depth displayed a higher mean $Q_{10}$ of 2.98, we estimated that net soil respiration might increase 1.16 folds more than $CO_2$ fixation rates under projected warming scenarios of 4 °C. The observed higher $^{13}C$ signatures in the SOC pool of the beech soil suggested higher microbial residue formation and this was reflected in the lower $Q_{10}$ values for $CO_2$ fixation rates per gram of microbial biomass for the beech than for the spruce soil. Also, the higher allocation of $CO_2$-derived carbon to the SOC pool at higher temperatures indicates that warming primarily results in an increased residue formation of microbial cells. Findings from this study indicate that dark $CO_2$ fixation in temperate forest soils might be less responsive to future warming than net respiration, and as a result, could recycle less $CO_2$ respired from temperate forest soils in the future than it does now.

**Data availability**

Raw data associated with this study can be accessed at https://doi.org/10.17617/3.EFHWIY (Akinyede et al., 2022b). Generated sequences obtained for all soil samples in this study are deposited in the NCBI Sequence Read Archive (SRA) database with accession numbers: SAMN26148471, SAMN26148472, SAMN26148473,

**Code availability**

Data analysis was done using only standard tests and plotting commands in R. These codes are available on request from the corresponding author.

**Author Contribution**

RA and KK planned the sampling campaign; RA carried out the campaign and performed the experiments and measurements; RA and MT analysed the data and wrote the manuscript draft. MS, ST, and KK reviewed and edited the manuscript.

**Competing Interests**

The authors declare no conflict of interest regarding this study

**Acknowledgements**

We thank Beate Michalzik and Florian Achilles for providing us with useful information on the soil properties and the history of the study site. We are grateful to Marco Pöhlmann, Jens Wurlitzer, and Stefan Riedel for soil sampling and incubation set-ups. We wish to acknowledge the contributions of Iris Kuhlmann for her support with CFE extractions, Armin Jordan for helping with gas measurements, and members of the routine measurements and analytical departments of the Max-Planck Institute for Biogeochemistry, Jena.

**Financial support**

This study was jointly supported by the Max Planck Institute for Biogeochemistry Jena (MPI BGC), the International Max Planck Research School for Global Biogeochemical Cycles Jena (IMPRS-gBGC), Deutscher Akademischer Austauschdienst (DAAD), and the Collaborative Research Centre 1076 AquaDiva (CRC AquaDiva), Germany. Funding by the DFG under Germany's Excellence Strategy - EXC 2051 - Project-ID 390713860 is also acknowledged.

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
