# Peer review of "Temperature sensitivity of dark CO2 fixation in temperate forest soils"

_Biogeosciences, 2022_

## Author Response (AR1)

*Dear Dr. Akihiko Ito*

*We would like to thank you and the reviewers for the helpful and insightful comments to improve our manuscript, "bg-2022-90". We are glad that you agree that we have provided adequate responses to the referee comments and that our manuscript would be reconsidered after major revisions based on the open discussion.*

*To meet the reviewers' concerns, we have revised the manuscript addressing all points raised during the open discussion and provided a point-by-point response below (see italicized texts). These include addressing the question on why the temperature sensitivity of dark $CO_2$ fixation differs for rates reported per gram of soil and per gram of microbial biomass carbon and explaining the rationale behind the quantification of the carbon allocation. We also included more information on the role of microbial communities in dark $CO_2$ fixation in the introduction, elaborated on the gene abundance results in the discussion and toned-down information on the effect of clay content on microbial biomass turnover. We in addition, clarified all confusing wordings and texts in the discussion, and on the sampling design and statistical analysis done and removed Figure 7. These are included as marked-up changes in the manuscript. We trust that this revised version of the manuscript would be suitable for publication in Biogeosciences.*

*Sincerely,*
*Rachael Akinyede (contact author)/Kirsten Küsel (corresponding author)*
*-on behalf of all co-authors*

**Editor's Comments to the author:**

Dear authors:
Thank you for making a fruitful discussion with the reviewers. Both referees gave positive evaluations on your manuscript; especially, referee #2 thought it has excellent presentation quality. I studied the discussion and conclude that you provide adequate responses to the referee comments. About your rebuttal about "C allocation" in Section 3.3, I asked the referee #1 about requirement of further clarification, and got no additional request. Therefore, my recommendation is that the manuscript would be reconsidered after major revisions on the basis of the open discussion.

**Reviewer 1: Comments to Author:**

This study explores the temperature sensitivity of microbial non-phototrophic CO2 fixation in temperate forest soils. The manuscript is interesting but some aspects are not clear and require improvement. Particularly, the authors should explain why the temperature sensitivity of CO2 fixation differs depending on whether it is reported per soil mass or per microbial biomass C (see below).

Main comments

1. Figure 2 Why does Q10 for CO2 fixation differ between the rates per MBC and soil? I assume this is due to differences in the MBC in the two soil subsamples that have

been exposed to different temperatures. It is rather surprising that the MBC concentrations differ so strongly, and it would be good to see the values (in a table).

*The difference in $Q_{10}$ based on rates per MBC and soil dry weight is not caused by differences in the MBC content as the MBC content were similar between the beech and spruce soils with hardly any changes with temperature (see Table S2, supplementary information). Instead, it is caused by the difference of the rates being assessed. For calculating $CO_2$ fixation rates per gram dry soil, we measured the excess $^{13}C$ in the soils and for rates per gram microbial biomass, we measured excess $^{13}C$ extracted in MBC (lines 209 - 222). Hence, we are looking at two different uptake rates: the rate of incorporation in total soil carbon (living and dead microbial biomass plus soil organic matter), and the rate of incorporation of the label into microbial biomass (lysable cells). The $Q_{10}$ values for both rates would be the same, if, for example, all $^{13}C$ in the soils was still in the living biomass at 4 and 14 °C.*

*However, for the beech soil, we found 23% more $^{13}C$ label incorporated into the MBC pool at 14 °C than at 4°C. In contrast, we found a 70 - 90% increase in $^{13}C$ label incorporation at the higher temperature in the SOC pool. This led to larger calculated temperature dependence of fixation rates expressed per gram of soil and hence, higher $Q_{10}$ values, compared to those calculated for rates expressed per gram MBC. This information has been clarified in the manuscript (lines 344 - 346, 355 - 357; 376 - 378; 521 - 529).*

2. Section 3.3 The rationale behind the quantification of the "C allocation" is not clear. Given that the incubation lasted only a few days, it is unrealistic that a lot of the microbial biomass C already turned into microbial necromass during the incubation. Thus, what the authors report here is probably rather the result of differences in the efficiency of the chloroform fumigation.

*We disagree with the reviewer that significant amounts of microbial biomass carbon cannot be transferred to SOM within the 21-day time frame of our experiment based on the different rates we measured. Our results show differences in the proportion of fixed $^{13}C$ in the SOC and MBC pools for the beech and spruce soils (see reply to comment 1). Since $CO_2$ fixation is a microbial process, we assume that the excess $^{13}C$ label measured in the soils after 21 days originates from $CO_2$ fixed by microbial biomass which has been transferred as microbial residues into the soil. Other studies found that microbial biomass can turn over quite rapidly as fast as 18 – 33 days in soils (Cheng, 2009; Spohn et al., 2016) and the transfer of microbial residues (both as turnover of necromass and formation of extracellular products from living biomass) into SOM can occur in as little as hours (Geyer et al., 2020) This information is now given in the manuscript (lines 524 - 538). Since we don't just expect microbial turnover via necromass production but also transfer of extracellular metabolites from living biomass, we have introduced the broader term, "microbial residues" everywhere it applies in the manuscript (lines 52 - 55; 316 - 318; 524 - 538; 539 - 542; 582 - 589). This term is defined as any non-living organic material of microbial origin including necromass and extracellular metabolites (Geyer at al., 2020).*

*The reviewer is correct that differences in CFE efficiency might affect the calculated carbon allocated to MBC due to possible effects on MBC measurements. However, previous studies using a $K_{EC}$ of 0.45 to account for the CFE extraction efficiency (Wu et al., 1990; Joergensen et al., 2011) as used in our study (lines 158 - 165), show that this factor does not strongly*

*vary between soils irrespective of differences in soil properties (Martens, 1995; Joergensen et al., 2011) and would not be incubation-temperature dependent. Hence, we argue that the difference in turnover described now as residue formation between beech and spruce soils and also with temperature, are caused by factors differentially affecting either the lifespan of microbial cells or the formation of microbial residues and this has been clarified in the discussion (lines 539 - 560). Additionally, the relationships of rates to MBC previously found in other soils (Akinyede et al., 2020; 2022a) suggest that the CFE efficiency does not differ dramatically between the two soils. However, we cannot exclude possible effects resulting from differences in CFE extraction efficiency on our results. We have thus added a sentence to the effect that: While previous studies do not show that the CFE extraction efficiency factor of 0.45 varies strong between soils or temperatures of incubation, the assumption that this is constant between depths may affect our results, especially in comparisons of rates between different soil depths, or between rates expressed per MBC and per gram soil (lines 166 -169).*

3. Lines 459-461 I agree with this sentence. In addition, the authors should also mention that changes in primary production and root exudation might completely change the response of the studied processes to changes in soil temperature, which adds further to the uncertainty to the predictions. Given these uncertainties, I strongly recommend to remove Fig. 7 from the manuscript.

*We agree and have included this statement in the manuscript (in line 496 - 502), and also removed Figure 7.*

4. Section 2.4 For how long were the soils explored to the 13CO2?

*In this study, all soils were exposed to the $^{13}C$-labelled $CO_2$ for a period of 21 days. We have modified the manuscript to clarify this (lines 200 - 201).*

Further comments

5. L. 25-27 Based on the determined parameters (respiration and CO2 fixation) no conclusion about microbial biomass turnover can be drawn

*In reference to our reply to comments 1 and 2, we still wish to speculate about the microbial biomass turnover which we have now described as a part of microbial residue formation. But following the suggestion from reviewer 2 comment 1, this speculation has been limited to the discussion section.*

6. L. 42 Remove "which also affects CO2 emissions from other soils"

*This has been modified as suggested (line 44).*

7. L. 52 Do you mean SOC concentration or quality?

*We meant both SOC content and quality. This has been modified in the text (line 62).*

8. L. 71 replace second "by" by "until"

*This has been modified as suggested (line 83).*

9. L. 77-80 This statement cannot be drawn from the cited studies since they measured both processes at only one temperature

*Thank you for this comment. Our assumption is not only based on the findings from past studies showing that dark $CO_2$ fixation rates correlate linearly with net soil respiration (Miltner et al., 2005; Santruckova et al., 2018). We also considered that both soil respiration and $CO_2$ fixation rates increase with temperature as stated in lines 69 - 71 and lines 77 - 80. This section has thus been modified accordingly (lines 89 - 93).*

10. L. 83 are there other forest types in the temperate zone besides coniferous and deciduous forests?

*For simplicity, we have rephrased the sentence in the manuscript to reflect deciduous and coniferous forests as the two temperate forest types based on vegetation as reported in past studies (lines 95 - 96).*

11. Table 1 Please indicate the depths of the soil horizons

*This table has been modified as suggested*

12. L. 441/442 Remove "derived"

*This has been modified as suggested (line 479).*

13. L. 462-464 These two sentences are not clear, at all.

*We apologise for the confusion; these sentences have been clarified (lines 503 - 506).*

14. L. 492-494 There seems to be some confusion here, and the process of microbial biomass turnover and microbial necromass turnover get mixed up. I think what the authors actually refer to is the rate at what C turns over in the living soil microbial biomass. It would be good to separate these two process more cautiously in the text.

*Thank you for this comment. We refer to the transfer or turnover of carbon from the living microbial biomass into the soil which we now describe as microbial residues (see reply to comment 2). Hence, we have removed the wording on necromass stability here in the manuscript (line 548).*

**Reviewer 2: Comments to Author**

The authors compared the temperature sensitivity of dark CO2 fixation and respiration in temperate forest soils of Germany. The fixed 13C was traced into microbial biomass and SOC, allowing the authors to comment on potential microbial biomass turnover rates under the contrasting temperature treatments. The study is interesting and the design is overall simple, but effective; there is limited information on the potential changes in dark CO2 fixation under climate change. However, several aspects of the analysis and the discussion could be further clarified. Some of the results pertaining to the microbial community response and gene abundance were not adequately addressed in the discussion.

1. Line 26: This speculation around the role of clay content may not be appropriate for the abstract. Since the role of texture was not directly studied it is best not to highlight this as a possible mechanism in the abstract. Many other aspects of the systems may be able to explain differences in microbial biomass turnover. Similar comment for the

last sentence of the abstract – "…variations in site-specific parameters might affect microbial biomass…"

*We agree and have limited our speculation on the role of clay content in microbial biomass turnover which we now describe as microbial residue formation (see reviewer 1 comment 2), to the discussion, and have removed this information from the abstract (lines 27 - 28; 33 - 35).*

2. Line 41-42: Add "through" so that it reads "through so-called dark CO2 fixation…". Also, what is meant by "which also affects CO2 emissions from other soils"? This wording is unclear.

*Thank you. The word "through" has been added to the sentence as suggested (line 44). For clarity and in line with our reply to reviewer 1 comment 6, we have removed the phrase, "which also affects $CO_2$ emissions from other soils" from the sentence (line 44).*

3. Line 66-67: What kind of ecosystems were included in this study by Nel and Cramer (2019)?

*This study was conducted for an afro-temperate forest and grassland ecosystems in Southern Africa, and this information has been included in the manuscript (line 78 - 79).*

4. Line 71: I suggest changing "can warm" to "projected to warm"

*This has been modified as suggested (lines 82).*

5. Line 77-80: Please add a sentence or two to discuss the potential reasons why these processes would be expected to mirror each other.

*Thank you for this comment. In line with our reply to reviewer 1 comment 9, we have modified this section and changed the wording "mirror" to clarify what we infer (lines 89 - 93).*

6. In general, the introduction could have more discussion of the microbial community's role in dark fixation.

*Following the reviewer's advice, we have included more information on the role of microbial communities in dark $CO_2$ fixation to the introduction (lines 52 - 59).*

7. Hypotheses appear to be implied in the writing, but could be explicitly outlined in this last paragraph of the introduction.

*Thank you for this comment. This has been modified as suggested (lines 100 - 101).*

8. Line 96-98: It is not clear what this is saying.

*We apologise for the confusion. Here we describe how the forest study site was established. We have modified the texts in this section to provide better clarity (lines 110 - 113).*

9. The beech and spruce plots were not replicated, correct? I am not sure it is possible to comment on statistical differences between spruce and beech plots without further replication of the forest types.

*We did not replicate each forest plot. However, during sampling, we took six replicate soil cores each from the beech and spruce plot (line 135). We later refer to these plots as "soil" and not "plot" in the manuscript and this number of soil core replicates are sufficient to compare these soils. For this reason, we have toned down our wording in the discussion on what our results might mean for other beech and spruce dominated soils (lines 599 - 602).*

10. Line 126: What is a "biological replicate"?

*By using the term biological replicate, we refer to the replicates of the soil cores taken from each sample plot (either the beech or the spruce plot) and this is to account for variability within each of the sampled plots. However, for simplicity, we have now only used term "replicate cores" in the manuscript (lines 143).*

11. Line 183: Should 12C/13C be 13C/12C?

*We apologise for the oversight. This has been corrected (lines, 209, 213).*

12. Line 279: What is the covariate in the ANCOVA?

*Thank you for this comment. When comparing some of the measured parameters between beech and spruce soil with ANCOVA, we included data from the whole soil profile for the beech and spruce soil. As a result, soil depth would also account for variability in these parameters. To account for this, soil depth was used as a covariate in the analysis. This has been clarified in the manuscript (lines 309 - 310).*

13. Line 504-506: The turnover may be slower in the clay-rich soil, but there is a greater availability of mineral surfaces that could potentially interact with C.

*It was not entirely clear to us what the reviewer's point is here. The spruce soil with higher clay content had slower biomass turnover or more accurately, lower residue formation compared to the beech soil (lines 389 - 401; 521 - 529). The original discussion focused mostly on the potential interactions with clay that could slow the transfer of residues from microbial biomass to SOM in the higher clay spruce soil, including the idea that more mineral surface area could also contribute to this (lines 539 - 560). We agree that clay can act in additional ways to explain the reduced transfer of label from microbial residues. For example, association with clay surfaces can lower residue formation in SOC if the production of microbial residues would be diluted by a larger overall inventory created by the higher clay content. Thus, the amount of label transferred would be more diluted resulting in a smaller proportion of total SOC. However, it is difficult to say the exact mechanisms for the patterns we observed especially since many parameters were not measured or tested experimentally. Nonetheless, we have added a respective sentence in the revised version (lines 560 - 563).*

14. In general, the authors should elaborate on the gene abundance results. There appears to be no comment on these results in the discussion.

*Following the reviewer's advice, we have provided more information about the gene abundance to the discussion section (lines 577 - 581).*